# An Italian Survey and Focus Groups on Fibromyalgia Impairment: Impact on Work and Possible Reasonable Accommodations

**DOI:** 10.3390/healthcare12020216

**Published:** 2024-01-16

**Authors:** Michael Tenti, William Raffaeli, Mery Paroli, Gabriele Gamberi, Riccardo Vincis, Barbara Suzzi, Corrado Fagnani, Laura Camoni, Virgilia Toccaceli

**Affiliations:** 1ISAL Foundation, Institute for Research on Pain, 47921 Rimini, Italy; william.raffaeli@fondazioneisal.it; 2Anesthesiology and Pain Therapy Unit, Santa Chiara University Hospital, 56126 Pisa, Italy; meryparoli@hotmail.it; 3ASPHI Foundation Onlus, 40126 Bologna, Italy; ggamberi@asphi.it (G.G.); riccardo.vincis@gmail.com (R.V.); 4Comitato Fibromialgici Uniti (CFU) Odv, 40055 Castenaso (Bologna), Italy; barbara.suzzi@gmail.com; 5Centre of Reference for Behavioral Sciences and Mental Health, Istituto Superiore di Sanità (Italian National Institute of Health), 00161 Rome, Italy; corrado.fagnani@iss.it (C.F.); laura.camoni@iss.it (L.C.); virgilia.toccaceli@iss.it (V.T.)

**Keywords:** fibromyalgia, work, disability, employment, focus group, reasonable accommodations, cross-sectional survey

## Abstract

Fibromyalgia symptoms affect the sufferers’ working life; however, through reasonable accommodations in workplaces, they can continue to work satisfactorily. There are no Italian studies on factors that facilitate or hinder fibromyalgia-affected people’s working life. Our objective was to explore, in a pre-pandemic setting, the quality of working life of fibromyalgia sufferers and reasonable accommodations to improve it. Quantitative and qualitative methods were applied; a survey-questionnaire, participatory-developed, was online-administered to a sample of self-reported FM sufferers (N = 1176). Then, two Focus Groups (FGs), involving 15 fibromyalgia-affected women, were held. Data were analyzed by a thematic analysis approach. Among survey-respondents, 20% were unemployed and only 14% went to work gladly. Variability of pain (84%) and fatigue (90%) were the most perceived reasons for difficulties at work. Negative relationships at work were reported by most participants. The FGs’ discussions addressed different strategies for overcoming the main obstacle of “not being believed by colleagues and employers” and reasonable accommodations. However, a negative hopeless attitude towards the solution of problems at work was also apparent. Different critical issues in the workplace emerged from the survey and the FGs. Coordinated actions, according to a transdisciplinary approach, are needed to manage fibromyalgia-induced difficulties in the workplace.

## 1. Introduction

Fibromyalgia (FM) is a chronic pain syndrome [1] affecting 0.2–6.6% of the general population, especially working-aged women, despite the lack of standardized tools of investigation, which may influence these estimates [2]. FM is characterized by widespread pain, fatigue, cognitive dysfunctions, sleep disturbances, and several other symptoms (e.g., stiffness, dyspepsia, genitourinary disorders) and comorbidities (e.g., irritable bowel syndrome, anxiety and depressive disorders, rheumatic disease) [3,4]. Decisive diagnostic tests and treatments are not yet available; consequently, delays and mistakes in the diagnostic process and difficulties in managing the syndrome frequently occur [5]. No specific biomarker exists for FM syndrome [6], and conventional analgesics and pain-modulating drugs (e.g., antidepressants, antiepileptics) are usually not very effective. Therefore, different guidelines for the management of FM syndrome highlight the importance of physical therapy and graded physical exercise, even in combination with other recommended non-pharmacological interventions such as cognitive-behavioral therapy [7,8]. Manual therapy is also providing promising results in the management of FM symptoms [9]. Diagnostic uncertainty, an unremitting course, symptoms’ heterogeneity, and treatment difficulties [5] seriously affect sufferers’ quality of life, particularly at work [10]. FM patients frequently develop a severe disability, which may prevent them from seeking, continuing, or resuming employment [11]. As recently reviewed by various research, 35–50% of FM patients do not work [12], and working ability is seriously compromised in a few years from the onset [13,14].

Work difficulties lead to a worse health status and psychological well-being, as well as reduced work productivity, early retirements, and financial compensations [12,15,16]. Pain and fatigue are the symptoms causing the most severe work difficulties [12], while FM severity has been found to be associated with a reduced job productivity [16]. However, besides symptoms severity, work disability may also significantly correlate with physical demands of the workplace. In fact, the interaction between physical limitations and job requirements is as important as age and comorbidity in explaining the high rates of work disability among persons with musculoskeletal conditions [17]. A recent French survey uncovered that risk factors for sick leave in FM women are related to the professional context rather than FM characteristics [18].

It is well known that work ability results from a balance between work demands and personal resources [19]). According to the International Classification of Functioning, Disability, and Health (ICF) [20], disability occurs only when external conditions are an obstacle to the person. The Convention on the Right of Persons with Disabilities [21] explicitly embeds the concept of “reasonable accommodation”, referring to necessary and appropriate adjustments, not imposing a disproportionate or undue burden for workplaces, which ensure people with disabilities can exercise their rights on an equal basis with others. Of note, when reasonable accommodations in workplaces are made and FM workers find an environment matching their ability, they can continue to work with satisfaction [10,14].

To the best of our knowledge, there are no Italian studies concerning factors perceived by FM-affected people as facilitating or hindering their opportunities to remain at work or reasonable accommodations that are able to fully enhance their working life. The underlying hypothesis is that FM-affected people in Italy, as already detected in other countries [10,11,12,13,14,15,18], might suffer from working difficulties, which require specific accommodations. Moreover, the gathered knowledge might uncover opinions and ideas shaped by social and cultural environments on what is considered undisputed or controversial for improving FM people’s life at work. The overall goal of the present study is, then, to explore barriers and facilitators that influence working well-being, the quality of relationships, communication within the organization, and other significant aspects of the professional life of FM workers in Italy. Furthermore, the study aims at identifying, from the point of view of FM-affected people, reasonable accommodations for improving their working life and better understanding how to handle and overcome FM workers’ difficulties.

## 2. Materials and Methods

A formal approval by an Institutional Review Board was not requested, as the study was a descriptive, non-experimental one, and the approval was not compulsory. However, as the ethics are a tenet for the research group, the study procedures were coordinated by a co-author (VT) with a background in bioethics to fully comply with the requirements of the Helsinki Declaration [22] and to give a privacy-by-design framework to the work.

Quantitative and qualitative methods were applied. The research is composed of two different studies. First of all, a cross-sectional survey was conducted through a questionnaire that was participatory-developed and online-administered to a convenient sample of self-reported FM sufferers. Then, Focus Groups (FGs) were designed for an in-depth study of a few survey findings.

For both investigations, potential participants had to be at least 18 years old (the age of majority). For what concerns the FG study, they had to be no older than 75, mainly because of the difficulties that FM sufferers may encounter in reaching the venue of the FG and sustaining a two-hour discussion, which can be exacerbated by the old age. For both studies, the main inclusion criterion was a previous diagnosis of Fibromyalgia; FG investigations had additional inclusion criteria (see the following sub-sections).

### 2.1. Online Survey

Potential participants were invited online, they were not known to the research group, and the survey was conducted anonymously. An invitation was sent by means of the Facebook page and the website of the CFU-Italia Odv, an association of fibromyalgia patients in Italy. Every patient/CFU-Italia member could see the invitation by entering the Facebook page or the CFU-Italia website, and after reading about the study’s aim, procedures, and burden, they could decide whether to participate or not. Therefore, according to this approach, potential participants had already received a diagnosis of FM, and they self-reported it by responding to a specific item in the questionnaire (see Figure 1). The survey was conducted between 4 June 2019 and 1 August 2019. The questionnaire was developed in Microsoft Form. It was designed by a panel of experts: a pain clinician (WR), a clinical psychologist (MT), an organizational psychologist (RV), and a disability manager (GG). Five representatives of CFU-Italia Odv, an Italian FM patients’ association, also took part in the questionnaire development to better align the process and outcomes of the research with FM sufferers’ needs. In particular, a few experts’ consultations allowed for defining the most relevant areas to investigate. Then, the experts’ panel defined the items to explore each area. The final version of the questionnaire was assessed for clarity, relevance, and completeness by CFU-Italia representatives.

The questionnaire included: gender, age, region of residence, housing conditions, years from diagnosis, employment status, and job position. Further items investigated working well-being, FM induced difficulties and communication with others, the quality of the relationship with colleagues and employers, and workplace barriers and facilitators. The questionnaire is available in Appendix A.

### 2.2. Focus Group Study

FG methodology was used to clarify the highly heterogeneous and often unfocused answers to the open questions about “Barriers and facilitators” in the online survey. In particular, FGs investigated: (i) perceived barriers and facilitators in workplaces; and (ii) possible solutions for the barrier of “not being believed” in the workplace, frequently addressed by the survey respondents. People aged 18–75 years, with a diagnosis of FM from a rheumatologist or pain physician, who were able to travel to the venue of the FG sessions and able to sustain a two-hour discussion, were included. People with hearing impairments or difficulties in understanding and/or using the Italian language or presenting acute psychiatric symptoms were excluded.

In September 2019, the recruitment started through the launch of dedicated posts on the Facebook page and groups of CFU-Italia Odv, addressing FM people who had already participated in the survey and had declared to have experienced difficulties in the work environment. Thirty-two FM sufferers communicated their interest in participating via email, and fifteen were admitted after an interview with a clinical psychologist (M.T.) who verified the satisfaction of the eligibility criteria (see Figure 1). After an informed consent procedure, they received a questionnaire on socio-demographic characteristics, FM diagnosis, and comorbidities, to be completed in October 2019.

The first FG took place in Bologna in November 2019, and the second took place in Rome in December 2019. A psychologist (MT) and a facilitator (FP in the first FG, SS in the second) conducted the FGs of two hours each, following a list of questions defined by the same experts’ panel that developed the questionnaire, included in Appendix A. The FGs were audio- and video-recorded and transcribed verbatim to analyze emerging themes. No further FGs were organized, since a large consistency was observed between the themes that emerged from the two FGs.

### 2.3. Ethical Issues and Personal Data Treatment

The study follows the international guidelines for the ethical conduct of research with human beings (Helsinki Declaration) [22] and complies with the legal norms for personal data protection (Reg EU 2016/679; Italian Legislative decree 196/2003). The questionnaire survey was conducted anonymously, and informed consent procedures were used for the FGs participation. Transcriptions were pseudonymized. After the time required for data quality control and analysis, all data were irretrievably anonymized, and video recordings were destroyed.

### 2.4. Data Analysis

Microsoft Excel 2017 was used for the descriptive analysis of the survey data. The thematic analysis [23] and the key questions asked during the FGs were used to analyze the FG, following a deductive approach. FG participants’ answers were also analyzed inductively to detect other possible themes and/or subthemes of significance. The FG analysis was conducted independently by MT and MP. Themes and sub-themes were revised by a third reviewer (VT) who previously read the transcripts. Two face-to-face meetings between MT, MP, and VT were, finally, held to resolve any discrepancies or overlaps. This study followed the consolidated criteria for reporting qualitative research (COREQ) guidelines [24] (Appendix A).

## 3. Results

### 3.1. Survey Results

Overall, 1217 subjects filled in the online questionnaire; of them, 41 were excluded because they did not answer the items on age and/or on time from FM diagnosis. Of the 1176 people with FM who responded to the survey, 94.3% were females; their mean age (±SD) was 46.8 years (±9.3) (age range: 18–75). The respondents were from all over Italy, and most of them (46.6%) declared to have received an FM diagnosis between 1 and 5 years before the survey. The participants’ socio-demographic, clinical, and occupational characteristics are presented in Table 1 and Table 2. Among the respondents (N = 1162), 45.7% were employees (21.8% of them teachers) and 41.5% were workers (36.3% in healthcare sectors).

The number of respondents to the following questions is 1163.

#### 3.1.1. Attitudes and Difficulties Experienced at Work

Only 14% of participants reported they go/went to work gladly and with satisfaction, while 53% go/went gladly despite some significant difficulties. A total of 17% manifested concerns for losing their job for FM-related reasons, and 16% do/did not like going to work. Pain variability (84%) and chronic fatigue (90%) were the most perceived reasons for the difficulties experienced at work; lower percentages were reported for mood issues (47%), a lack of lucidity due to medications (36%), and negative interpersonal relationships (24%).

#### 3.1.2. Characteristics of the Relationships Experienced at Work

Negative relationships in one’s work environment, characterized by isolation, disqualification, or even open hostility, were reported by the vast majority of participants, both regarding colleagues (60.1%) and employers (70.5%) (Table 3).

Only 19.2% of respondents declared that people in their work environment know FM and what it entails, while 46.1% stated they felt to be believed when they expressed their difficulties. Table 4 reports how FM workers feel and whom they communicate with in the work environment when experiencing FM-related difficulties or medications assumptions.

#### 3.1.3. Difficulties in the Workplace

Figure 2 reports difficulties (and related frequencies) FM workers experienced in the work environment due to FM symptoms and whether they were solved or not. More than half (53.9%) of the FM workers experienced difficulties related to the work organization, the division of tasks, or tasks being assigned several times, and they have always solved them.

A high proportion of the FM workers (around 40%) have experienced other difficulties several times, and they never or almost never solved them. These difficulties were related to different organizational, practical, or career aspects of the work. In particular, they noted timetables (e.g., entry/exit, access to part-time work), the presence (e.g., repeated sick leaves), and work rhythms (e.g., breaks); the physical environment, premises, and structures (e.g., long distances to be covered on foot or by stairs); the specific rule of the company; equipment or technologies; and the chance of obtaining a promotion or improving one’s position.

Finally, more than half (60.7%) of the FM workers have experienced difficulties related to the workstation or seat (e.g., ergonomics) on several occasions and have never or hardly ever solved them.

#### 3.1.4. Barriers and Facilitators

Answers to the open questions regarding barriers and facilitators in the workplace showed high heterogeneity and often unfocused and unclear favoring or hindering factors. However, the most frequently suggested facilitators were related to: work environment conditions (e.g., proximity, accessibility); work flexibility (e.g., timetable flexibility, smart-working, work breaks, change of tasks, limiting weights); postural changes and ergonomic tools; minor pressures in terms of responsibilities, times, and error handling; psychological characteristics (e.g., optimism, determination, patience); the presence of supporting colleagues/collaborators; and being believed by colleagues and employers. Among the hindering factors there are: a high workload (a hectic pace and stress, an inability to plan work, too many responsibilities and deadlines, heavy lifting); work environment factors (e.g., distance from home, prolonged sitting or standing, work shifts, thermal shocks); psychological disturbances (e.g., comorbid anxiety or depression); and not being believed. “Being believed” and “not being believed” appear to be, respectively, the most facilitating and the most hindering factors reported.

### 3.2. Focus Group Results

The FGs involved a total of 15 female participants (8 in Bologna and 7 in Rome). The mean age (±SD) was 49.2 years (±9.1), (range 32–61 years). The majority of participants (53.3%) had a high school diploma, 26.7% had a university degree, 13.3% had a lower secondary school diploma, and 6.7% had a vocational diploma. Ten participants had a job, three were unemployed, one was a student, and another one was a retired person. On average (±SD), the participants declared they had been suffering from chronic pain for 16.9 (±11.6) years (range 5–40 years) and had received an FM diagnosis 6.1 (±4.3) years before (range 2–18 years). Almost all of them (N = 14) declared they had other comorbid pathologies, mostly osteoarthritis, small fiber neuropathy, and thyroid disease.

Table 5 synthesizes ex ante and inductively identified themes and sub-themes and verbatim quotes.

#### 3.2.1. How to Overcome the Obstacle of “Not Being Believed”

The need for more awareness-raising activities on FM emerged as a priority from both FGs. The participants suggested these activities should be directed to work and family environments, as sufferers often feel not understood or disbelieved either by colleagues, employers, or family members. As M. witnessed, “*Nobody believes me. Where I work, I have not told everyone that I suffer from Fibromyalgia, because some people would take the chance to fire me. However, I told someone and, according to them, but also according to my family, I am depressed, or lazy, even though I have been working since I was very young and raised 4 children alone*”.

However, awareness activities alone are considered to be not enough to overcome work difficulties. As B. said, “*Where information is not followed by tangible change actions, in the absence of protection, the fact that I receive work benefits for my health condition falls within the sphere of human relations. If I put myself in a position to be “attacked” [taking holidays, asking for more flexible hours, etc., Editor’s note], it’s easy for someone to slowly take my place and I eventually get fired for obvious reasons: because I’m sick and because I’m not efficient. So, information alone, what is it for?*”. All FGs participants agreed with the crucial importance of an “official” recognition of FM, namely, its inclusion within the Essential Level of Assistance (“ELA”) and disability tables that would determine a real certification of the existence of FM and its burden. Few participants suggested to manifest the needs of FM sufferers to policymakers (e.g., with flash-mobs or demonstrations) to make them aware of the problem’s seriousness.

The importance of patients’ associations was also underlined. Apart from their role in awareness-rising activities, they may also represent FM sufferers in front of institutions with a more cohesive voice.

A further strategy suggested for solving the problem of not being believed was the organization of trainings on pain and FM to General Practitioners, often at the forefront for FM diagnosis and treatment, and to Social Security doctors. E. said: “*I was home from work on sick leave. I went to clean the garden, I wasn’t even able to hold the broom in my hand, but at that moment the Social Security doctor arrived, he thought I wasn’t sick, since I was in the garden sweeping! A few days later another Social Security doctor came to visit me, who knew about Fibromyalgia, and in a similar situation, he didn’t flinch and recognized my need for rest!*”.

Obtaining a diagnosis of FM by a rheumatologist/pain physician may be a further facilitator for being believed. The diagnosis was considered to have a greater impact if sustained by a laboratory test. Participants, therefore, addressed the importance of scientific research on the neurobiological mechanisms underlying FM. Clear evidence would be fundamental to certificating the organic bases of FM and developing fast and accurate diagnostic tests.

There is a tendency to hide health issues and not talk about them to others, maybe out of the fear of being disbelieved or because the expression of personal difficulties may be a double-edged sword: “declaring” to suffer from FM can create discrimination and stigmatization as a depressed or lazy person. However, some others thought that not expressing one’s own difficulties may prevent others from knowing them. E. said: “*If I hide my problems, no one will ever know that I have certain difficulties. If I express my difficulties at work, at home etc., my environment begins to receive a certain nourishment*”.

Table 6 synthesizes strategies for overcoming the obstacle of “not being believed” proposed by the FGs participants.

#### 3.2.2. Reasonable Accommodations for Improving Work Environments

Many participants reported the importance of organizational aspects as facilitators that are able to improve the performance of FM workers. Among these, smart working represents one of the greatest facilitators. As reported by E., it would be very useful even for few days a week, perhaps when the person is in a bad health condition: “*In my case it should be useful even for two or three days a week. It would change my life, because I often have one day a week when I feel bad … I could avoid taking the whole sick day*”.

Other participants suggested the use of a greater flexibility in working hours, as this would allow the person a more flexible timetable in compliance with the agreed weekly hours. A. said: “*It would help me if I could be a few hours late at work; sure, as long as I come and do what I have to do. Because if I feel unwell one morning and am unable to move …*”.

The reduction in working hours, having more time to carry out tasks, frequent postural changes, and the avoidance of heavy duties were indicated by participants as further reasonable accommodations. Particularly, participants suggested adjusting the work tasks to the severity of FM and providing regular check-ups to monitor its progression. However, it was pointed out that these arrangements should not affect one’s salary or career advancement. They also suggested promoting a partial overlap of roles and tasks among colleagues so that other colleagues can temporarily take care of the FM worker duties. However, this was controversial due to the fear that the overlap of such roles and tasks could eliminate workers’ indispensability.

In the economic–legislative field, many suggested the importance of work licenses for medical visits or health treatments. E. addressed the need to modify some national collective labor agreements to guarantee sufficient hours of leave to workers, especially those suffering from chronic diseases. Other important benefits suggested were a reduction in the retirement age and the possibility of financial aid for hiring domestic workers (e.g., housekeepers, babysitters).

Both FGs, then, suggested numerous factors related to the physical environment, aids, technologies, personal protective equipment (PPE), and measures capable of improving FM workers’ functioning. It should be noted that the usefulness of one PPE or another, as well as the best adaptation of the working environment, appeared subjective. For example, the best microclimate for one person may not be suitable for another. V. explained: “*Fibromyalgia causes hypersensitivity, but, unfortunately, everyone has his/her own hypersensitivity, then everyone has his/her own needs*”. A few participants noted that many pieces of PPE and adjustments to the working environment are already provided for by the Italian D. Lgs 81/2008. Hence, there is a need for greater control by the competent bodies regarding the implementation of the norms in the company, as well as for reporting situations in which they are not respected. Some others pointed out the importance of economic incentives (e.g., tax exemptions) for companies and the development of social and welfare policies to provide facilities for health protection.

The presence of incentives for companies to ease the recruitment of FM workers could be a further facilitation to consider. P. reported: “*The question is, given that the company already has the minimum required number of employees with disabilities, why should they hire me instead of another healthy person at the same cost, given that I make less, and do I ask for one permit after another? There is no reason. Why, if there is no incentive, should they hire a person who has shortcomings?!*”.

According to all the participants, until FM is included in the ELA and disability tables, those who suffer from it are unlikely to be able to obtain reasonable facilities or accommodations in the workplace. C. said: “*I have always tried to adapt the working environment to my needs by asking questions for mobility, exchanges with other people etc. But now I work with colleagues who unfortunately have other recognized pathologies themselves and if one wants to put the weight of the colleague on the scale of the company compared to mine, that of the colleague will always be much greater. It becomes more complicated for me to demonstrate and assert my difficulties … especially if we consider that pain is not visible from the outside*”.

Finally, staff training on FM and assertive communication with colleagues could represent an important and reasonable accommodation to facilitate interpersonal relationships, often strained by the disease and its drawbacks.

Table 7 summarizes the reasonable accommodations proposed by the FGs participants.

During the FGs, the interactions were overall positive, despite the tendency of some participants to refocus their attention towards the possible negative implications of the proposed solutions (e.g., “Yes, but …”) and repeated digressions on the problems of treatment and FM’s impact on the quality of life. These digressions did not hinder the progress of the discussion, considering the numerous difficulties encountered by the participants themselves in finding solutions to the problems investigated, and are understandable if we consider that the participants were involved in a topic that concerns them closely.

## 4. Discussion

Our aims were to explore the quality of working life of FM people, barriers and facilitators that influence it, and possible reasonable accommodations. In our sample, the unemployment rate accounted for 20%. Although FM-affected people’s unemployment rates are generally high and not homogeneous, varying from 10% to 57% in Western countries [10], our rate is similar to that found in other recent Italian studies [25,26].

The survey highlighted a rather generalized difficulty in finding solutions to critical issues in the workplace, from ergonomics to the flexibility of working hours, physical environment-related factors, and career advancement chances. Barriers in the work environment, frequently reported in the online survey, were related to “not being believed”, a high workload (e.g., a hectic pace and stress), work environment factors (e.g., prolonged sitting or standing), and psychological disturbances (e.g., comorbid depression). These results are partially in line with those reported by Waylonis et al. [27] several years ago. In their study, in fact, activities reported as aggravating FM symptoms were computer activities or typing (37%), prolonged sitting (37%), prolonged standing and walking (27%), stress (21%), heavy lifting and bending (19%), and repeated moving and lifting (18%). Moreover, it is crucial to address the impact of the recent spread of COVID-19 on FM-affected individuals. Some studies have actually found that FM patients frequently reported a worsening of symptoms during the COVID-19 pandemic due to social distancing, economic issues, and difficulties in accessing medical and complementary treatments [28], while other studies have highlighted changes in the employment status of FM individuals during the COVID-19 pandemic. Particularly, an Israeli study conducted during the COVID-19 outbreak found that, among 233 FM participants, 11.7% were working less than usual, 21.6% were on unpaid leave, and 3.5% had been fired [29]. Conversely, another study suggested some beneficial changes in working habits, as smart working has improved well-being at work for FM individuals and even FM symptoms [30]. These findings make our results useful for future comparisons between pre-pandemic and post-pandemic evaluations of barriers and accommodations for FM subjects in the work environment.

FGs confirmed the major critical issues reported in our online survey, particularly for what concerns ergonomics. According to FGs, it is crucial to adopt ergonomic chairs, desks, and monitors as well as to ensure frequent postural changes. This is in line with the findings of a Swedish qualitative study on FM women that emphasized work posture and ergonomic aids as favorable factors for improving working condition [31]. Our FG participants were particularly sensitive to the importance of verifying the effective application of the current legislation concerning ergonomic criteria. Many of the solutions proposed are, in fact, already provided for by the current national legislation concerning health and safety in the workplace.

The FGs’ participants often reported smart working as an important facilitator for critical issues involving work organization and rhythms. Indeed, a recent Italian observational study conducted during the COVID-19 pandemic on a sample of FM patients [30] showed that smart working can permit to set up more flexible agendas and less stressful work routines—for example, by reducing the need for long daily trips and consequently increasing the time availability for other activities such as physical exercise.

Greater flexibility and a reduction in working hours could be further facilitating factors regarding continuing to work. Liedberg and Henriksson [31] already focused on how challenging it can be for a woman with such a demanding syndrome as FM to work full-time and take care of her family and herself.

The need for awareness-raising activities for FM in the workplace was also stressed. A lack of knowledge about FM symptoms and consequences seems to be perceived as one of the main reasons behind the attitudes of devaluation, distrust, or open hostility experienced in the relationships with managers and colleagues. Nonetheless, the FGs’ participants pointed out the importance of training activities in the workplace aimed at encouraging positive relationships between colleagues and employers, leveraging empathic skills and reciprocal respect. Interestingly, the literature already suggested that perceived empathy may play an important role in fostering positive interactions with coworkers and in employee well-being improvements [32], and the need for a change in attitudes and behavioral responses towards workers with disabilities has already been well highlighted by Liedberg and Henriksson [31].

It is also worth noting that when a problem related to FM arises, over 70% of sufferers prefer not to talk about it. This appears understandable, considering the difficulties that people with this syndrome often face, as their pain and difficulties are invalidated as if these were not real. Invalidation can also be accentuated by the sheltering role of “chronic pain” or other definitions used to refer to FM, which can generate doubts and confusion about the reality of symptoms [33]. Mukhida and colleagues [12] already highlighted that, given the symptoms’ invisibility, dislikes and misunderstandings easily arise in workplaces, producing stigmatization and reducing well-being for FM workers. Of note, one out of four survey respondents stated they had been devalued in their relationships with colleagues, and one out of three stated they had been devalued in their relationships with job managers/employers. Invalidation and disbelief are among the most critical factors for people with FM associated with symptoms of psychological distress, such as depression, guilt, and anger and feelings of isolation [34,35,36]. It is understandable how the fear of not being believed can lead to hiding one’s suffering to avoid invalidation, but in the long term this attitude can prevent people from obtaining attention and recognition. Indeed, the FG results emphasize the importance of expressing personal difficulties openly to help raise awareness in the living environment. At the same time, the expression of personal difficulties also produces fear of the stigma, which addresses more complex societal problems. A further controversy was noticeable between the need for an overlapping of roles and tasks in the work environment and the fear that this overlapping could be a hazard for the specificity and indispensability of FM-affected people. These controversial issues might also reflect a high degree of uncertainty and a lack of guarantees in the free labor market in the Italian context.

Finally, some FG participants were shown to have inaccurate information on FM, stating, for example, that it is a degenerative disease when there is no evidence in this regard or expressing divergent beliefs about the existence of accurate diagnostic tests. These observations address the importance of awareness-raising activities for FM, targeting not only civil society but also and primarily FM-affected patients themselves.

### 4.1. Clinical Implications

Our results may have different clinical implications. For what concerns the barrier of “not being believed”, psychoeducational interventions on FM syndrome in the work environment could improve colleagues’ and employers’ knowledge of FM, its symptoms, and the difficulties it entails. Interventions aimed at improving empathy and assertiveness skills might also be helpful for an amelioration of work relationships. The adoption of reasonable accommodations and a tailored adjustment of rhythms and the workload can have a further positive impact on the well-being of FM workers. Actually, there are few studies on the effectiveness of workplace and ergonomic interventions for FM symptoms. In a Randomized Controlled Trial, Martins and colleagues [37] found that an interdisciplinary program including ergonomics and postural orientations and approaches to occupational features improved the quality of life of FM patients. More generally, two systematic reviews found some low-quality evidence to support the effectiveness of ergonomic interventions (e.g., ergonomic chairs, ergonomic workstation redesign) in addressing the secondary prevention of musculoskeletal conditions [38]. Data from qualitative studies, however, suggest that reasonable accommodations and strategies for enhancing work ability and promoting positive changes may need to be tailored to the specific work environment and supported by healthcare professionals and employers [39,40].

### 4.2. Limitations and Future Directions

The present study has some limitations. We included “self-reported” FM patients; we do not actually know the extent to which our respondents accurately satisfied the ACR diagnostic criteria for FM [41]. Moreover, we do not know whether there could be distortion due to a response bias, as the characteristics of those who did not answer the survey are unknown. However, the sample obtained is large; just to give an idea of its size, it is comparable to 50% of the whole enlisted group of patients enrolled in the Italian Fibromyalgia Registry at the time of our survey, with similar socio-demographic characteristics [42]. Another limitation was the completely female sample. Even if FM is a relatively female syndrome, some authors have suggested the presence of a potential biased patient selection [43]. Therefore, efforts should be made in future studies to also involve male affected people and consider social and job-related gender issues.

Furthermore, it is important to point out that these results were obtained in a pre-COVID-19 pandemic setting. The worsening of symptoms described in the literature may have led to further difficulties in the workplace, which deserve additional investigations. In any case, our study may be considered a useful baseline for future studies and for a thorough assessment of any changes that might have occurred during and after the pandemic period for what concerns FM barriers and reasonable accommodations in the workplace.

Finally, as a next step, given the richness of information that the survey has gathered, an in-depth data analysis, based on multivariate statistical models and exploiting evidence from FGs, will be planned and implemented to investigate associations and mutual interactions between FM-relevant variables (e.g., quality of the relationships) while taking into account possible confounding factors (e.g., age).

## 5. Conclusions

The issues emerging from the present study are numerous and often intertwined. Low levels of perceived solidarity from colleagues and employers, a pretty high percentage of hostile behaviors recorded in the work environment, and a rather poor knowledge of the syndrome in itself, regarding both FM subjects and colleagues and employers, are among the main results of this national survey. The complexity of these results is strengthened by the high percentage of FM workers, nearly half of the whole sample, who have never solved practical, ergonomic, or organizational problems. The FGs also had the merit of highlighting that a few proposals regarding communication issues and the overlapping of job roles are really controversial facilitators. Related risks of stigmatization and losing a worker’s specificity are clearly linked to social, economic, and cultural issues at the national level. For these reasons, it is advisable that the implementation of possible actions to help FM workers express their full potential in the work environment is carried out according to a transdisciplinary approach. The coordinated contributions of disciplines such as medicine, psychology, sociology, economics, and patients’ associations consultations are crucial to this purpose.

## Figures and Tables

**Figure 1 healthcare-12-00216-f001:**
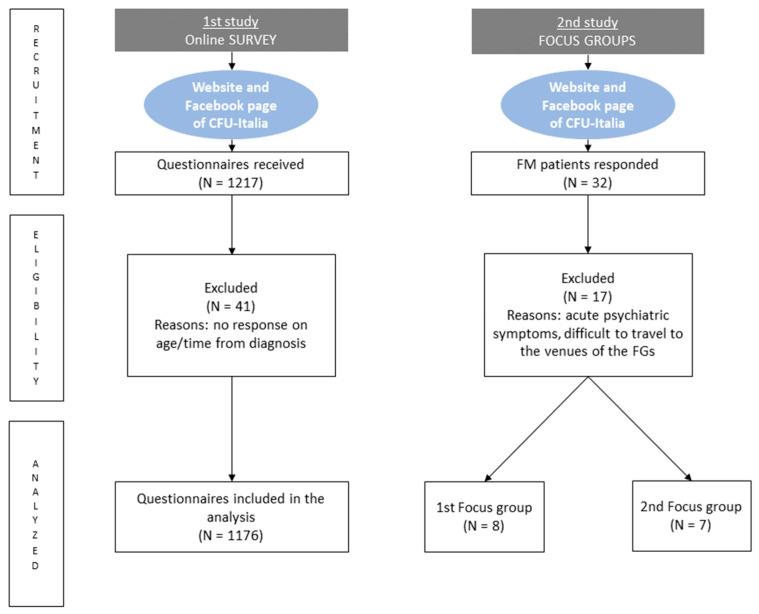
Flow chart of the recruitment for the two studies.

**Figure 2 healthcare-12-00216-f002:**
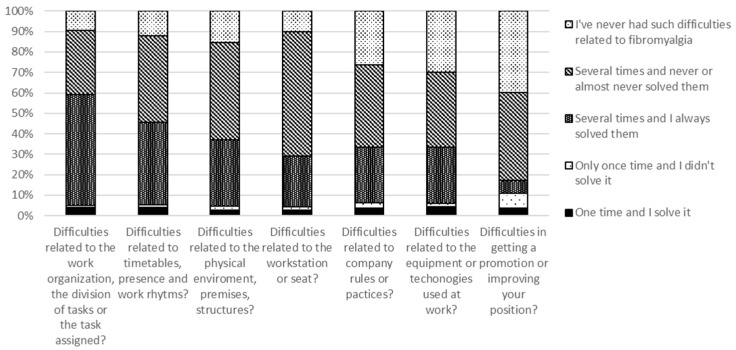
Difficulties that FM workers experienced in the work environment due to FM symptoms.

**Table 1 healthcare-12-00216-t001:** Main characteristics of the survey sample.

Variable	N	%
Age		
18–34	134	11.4
35–44	302	25.7
45–54	483	41
55–64	247	21
>65	10	0.9
Gender		
Female	1109	94.3
Male	67	5.7
Residence area		
Northern Italy	639	54.3
Central Italy	223	19.0
Southern Italy	314	26.7
Family situation		
Living with other people	1017	86.5
Living alone	159	13.5
Occupational status		
Employed	941	80
Unemployed	235	20
Years of unemployment		
<1 years	91	38.7
1–5 years	77	32.8
5–10 years	32	13.6
>10 years	22	9.4
Never worked	13	5.5
Years from FM diagnosis		
<1 years	170	14.5
1–5 years	548	46.6
5–10 years	246	20.9
>10 years	212	18

**Table 2 healthcare-12-00216-t002:** Job position and related working status among survey respondents (N = 1162).

Job Position	N (%)	Employed N (%)	Unemployed N (%)
Employee	531 (45.7)	457 (86.1)	74 (13.9)
Worker	482 (41.5)	379 (78.6)	103 (21.4)
Self-employed	77 (6.6)	58 (75.3)	19 (24.7)
Other	72 (6.2)	47 (65.3)	25 (34.7)

**Table 3 healthcare-12-00216-t003:** How do you value the relationships with your colleagues and with your job managers or employers?

Answers ^1^	Colleagues N (%)	Job Managers or Employers N (%)
I have their solidarity	180 (15%)	111 (9%)
They understand my health condition	399 (34%)	308 (26%)
They never involve me	275 (23%)	257 (22%)
I am devalued	294 (25%)	399 (34%)
There have been hostile behavior episodes	330 (28%)	418 (36%)

^1^ Multiple answers.

**Table 4 healthcare-12-00216-t004:** When I have difficulties related to FM or the medications I take.

Answers ^1^	N (%)
I prefer not to talk about it	706 (69.9)
I know who I can contact at work for help/support	198 (23.8)
I feel solidarity from my colleagues	208 (25)
Occupational doctors/Workers’ Health and Safety Representatives are important references	181 (21.9)
Company Unions are important references	78 (9.9)
The Employer/Personnel Manager/Department Head are important references	159 (19.6)

^1^ Multiple answers.

**Table 5 healthcare-12-00216-t005:** FGs themes and major sub-themes.

Theme	Subtheme	Verbatim
How to overcome the obstacle of “not being believed”	Strategies and facilitators	*“If I hide my problems, no one will ever know that I have certain difficulties. So, it’s important to talk about it openly.”* *“An official diagnosis from a specialized medical centre would certainly help!”*
How to improve work environments	Obstacles and barriers	*“A chaotic environment, with lots of noises, smells … it’s devastating”.* *“Even a great load of work and responsibilities does not help... responsibility gives anxiety and, unfortunately, I was able to handle it before the FM, now I can’t handle it anymore”.*
	Supports and facilitators	*“I asked my company to buy an ergonomic chair to be able to work many hours on the computer”.* *“Short but frequent breaks from work are essential for me”.*
	Application of already existing laws ^1^	*“For video terminal operators the law on safety in the workplace would be enough, it is very well done, but in Italy it is not applied”.* *“Breaks from activity would be important, and, theoretically, they are provided for by law!”.*
Lack of knowledge about FM ^1^		*“FM patients have a disabling and degenerative disease, so they should retire early”.* *A: “I have a disability because I have FM”. E: “Impossible, you’d be the first!”. A: “No, it’s not true, in Italy a disability for FM is recognized”. E: “I do not think so...”.*
Negative attitude towards solutions ^1^		*“We’re talking about science fiction!”* *“We are all giving indications that we know will not be feasible.”* *“Dedicated parking spaces would be useful, but they [the Company] will never give them to us!”*

^1^ Themes and sub-themes inductively defined.

**Table 6 healthcare-12-00216-t006:** Strategies proposed for overcoming the obstacle of “not being believed”.

Strategies	Questionable
Awareness and information activities aimed at civil society	No
Social security FM recognition	No
Awareness-raising activities	No
Representation activities by patient associations	No
Health training	No
Diagnosis from a specialized center	No
Communicating one’s difficulties due to FM	Pros: sensitization action. Cons: risk of stigmatization.
Biomedical research improvements	No

**Table 7 healthcare-12-00216-t007:** Reasonable accommodations proposed during FGs.

Type	Accommodations	Questionable
Organizational aspects of work	Smart working	No
	Greater flexibility in working hours	No
	Working hours reduction (with the same salary)	No
	More time availability for working tasks	No
	Avoidance of harmful activities (change in duties)	No
	Duties adjustment based on FM severity	No
	Work permits	No
	Partial overlapping of roles	Pros: tasks’ overload reduction. Cons: worker’s indispensability threatened
Economic and legislative field initiatives	Incentives for companies hiring FM-affected people	No
	Staff training on FM	No
	More controls on the application of the regulations	No
	Economic benefits for companies wishing to comply with regulations	No
	Subsidized health, social, and welfare policies development	No
	National Collective Labor Agreements amendments	No
	Retirement age reduction	No
	Economic aid for hiring domestic workers	No
Work environmental and equipment accommodations	Specific PPE for hearing, respiratory tracts, sight, smell, etc.	No
	Reserved parking spaces	No
	Ergonomic chairs and desks and anti-glare/blue light-shielded monitors	No
	Lecterns	No
	Safety shoes	No
	Regular breaks (especially from PC work)	No
	Frequent postural changes	No
	Microclimatic factors and air conditioning adjustments	No

## Data Availability

The data presented in this study are available on request from the corresponding author.

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
