# Peer review of "An Italian Survey and Focus Groups on Fibromyalgia Impairment: Impact on Work and Possible Reasonable Accommodations"

_healthcare, 2024, doi:10.3390/healthcare12020216_

Round 1

Reviewer 1 Report

Comments and Suggestions for Authors

The authors have produced an interesting and useful study whose objective was to identify, from the point of view of people affected by fibromyalgia, reasonable adjustments to improve their work life and to better understand how to manage and overcome the difficulties of workers with fibromyalgia.

However, I would like to make some observations before recommending your work for publication.

1. Could the authors add the type of study in the title?

2. The analysis performed with excel is interesting, but I think that the R program offers greater possibilities. Can the authors justify this choice?

3. I recommend the authors to comment very briefly on the therapeutic approach to finromyalgia. As a conservative treatment, there is a recent and novel study using different methods of manual therapy: DOI: 10.3390/ijerph20021061

4. To further enrich the authors' work, I recommend discussing how the confinement and pandemic situation due to COVID-19 has affected fibromyalgia patients: DOI: 10.29333/ejgm/11798

5. In the Discussion section, could you add a section on "Clinical Implications"?

6. Authors are requested to add a section on "Limitations and Future Directions" at the end of the Discussion section.

Comments on the Quality of English Language

No comments

Author Response

Reviewer 1

The authors have produced an interesting and useful study whose objective was to identify, from the point of view of people affected by fibromyalgia, reasonable adjustments to improve their work life and to better understand how to manage and overcome the difficulties of workers with fibromyalgia.

However, I would like to make some observations before recommending your work for publication.

  1. Could the authors add the type of study in the title?

ANSWER: The type of study has been specified within the title.

  1. The analysis performed with excel is interesting, but I think that the R program offers greater possibilities. Can the authors justify this choice?

ANSWER: We agree that R software offers greater possibilities than MS Excel. However, as the purpose of our investigation was, as a first step, purely descriptive and no formal statistical tests were performed at this stage, we have considered that MS Excel was adequate.

  1. I recommend the authors to comment very briefly on the therapeutic approach to finromyalgia. As a conservative treatment, there is a recent and novel study using different methods of manual therapy: DOI: 10.3390/ijerph20021061

ANSWER: We thank the reviewer for this comment, which allowed us to describe aspects of the syndrome that we had not yet commented. We added some information about diagnostic and treatment difficulties of FM syndrome (especially with analgesics) in the Introduction section. We then reported the importance attributed by some guidelines to non-pharmacological strategies. Finally, we indicated manual therapies as a promising intervention for fibromyalgia symptoms management (see lines: 48-49).

  1. To further enrich the authors' work, I recommend discussing how the confinement and pandemic situation due to COVID-19 has affected fibromyalgia patients: DOI: 10.29333/ejgm/11798

ANSWER: We thank the reviewer for this comment, which helped us to improve our discussion. Even though we cannot really discuss how the pandemic affected the work experience and clinical conditions of the patients in our study, as our research was conducted just before the Covid-19 pandemic, we have reported in “Discussion” (lines 385-397) the results of a few studies investigating the impact of Covid-19 outbreak on fibromyalgia patients, and we have highlighted how these studies suggest the need for further investigations in the post-pandemic setting. Moreover, in “Limitations and future directions” (a new section, suggested by the reviewer), we have addressed the usefulness of providing, as in our study, a description of pre-pandemic scenarios as a basis for future comparisons (see lines 487-490).

  1. In the Discussion section, could you add a section on "Clinical Implications"?

ANSWER: According to the reviewer’s request, we added a new section addressing “Clinical Implications” within the Discussion.

  1. Authors are requested to add a section on "Limitations and Future Directions" at the end of the Discussion section.

ANSWER: According to the reviewer’s request, we added a new section addressing limitations and future directions after the Discussion.

Reviewer 2 Report

Comments and Suggestions for Authors

This study is intriguing, particularly due to its extensive participant inclusion. I offer a few minor suggestions:

  1. Consider merging the sentence between lines 45-47 with the following paragraph.
  2. Place the study hypothesis at the end of the introduction for clarity.
  3. It is advisable to provide Institutional Review Board (IRB) approval details in the initial section of the methods.
  4. Is there a specified age range for the participants?
  5. In terms of data analysis, could you specify if any statistical tests were employed for comparisons?

Author Response

Reviewer 2

This study is intriguing, particularly due to its extensive participant inclusion. I offer a few minor suggestions:

  1. consider merging the sentence between lines 45-47 with the following paragraph.

ANSWER: Owing to further lines added, the sentence at lines 45-47 has been shifted and then merged with the subsequent paragraph (see lines 55-57).

  1. Place the study hypothesis at the end of the introduction for clarity.

ANSWER: The hypothesis has been explained in Introduction (see lines 77-82).

  1. It is advisable to provide Institutional Review Board (IRB) approval details in the initial section of the methods.

ANSWER: A short paragraph has been moved from that already present at the end of the manuscript (sub-heading “Institutional Review Board Statement”) to the initial section of Methods (see lines 90-95).  Moreover, actions and procedures adopted to guarantee the respect of the Helsinki Declaration are explained in details in the Materials and Methods section, sub-heading “Ethical issues and personal data treatment”.

  1. Is there a specified age range for the participants?

ANSWER: Potential participants had to be at least 18 yrs to participate. No methodological reasons or hypotheses have limited the age range of participants in the survey, thus also hampering the (already large) sample size. However, considering the difficulties generally encountered by fibromyalgia-affected individuals, for the focus group study we preferred to limit participation to those not older than 75, to avoid the risk of exacerbating their potential difficulties. We have better specified the age of participants in the Materials and Methods section (see lines: 101-106).

  1. In terms of data analysis, could you specify if any statistical tests were employed for comparisons?

ANSWER: No statistical tests were used to make comparisons; our study had a purely descriptive purpose as a first contribution, at national level, regarding “work experience and reasonable accommodations for FM-affected people”. However, our results may stimulate future research hypotheses about specific between-variable associations (e.g., “Are the quality of relationships in the workplace and the impact of fibromyalgia associated?”), or between-group comparisons (e.g., “Are there any differences in the perceived quality of relationships in the workplace between FM workers who go to work gladly and those who do not?”). For these future purposes, multivariate statistical models simultaneously incorporating the effect of the targeted explanatory variables with their mutual interactions, as well as relevant confounding factors, will be needed.

Reviewer 3 Report

Comments and Suggestions for Authors

The manuscript describes a quantitative and qualitative assessment of patients suffering from fibromyalgia in relation to their work environment.

Introduction

Well referenced but perhaps some mention could be made of previous studies that assess the interventions performed for these patients in the work environment, as well as the other symptoms that these patients also report besides fatigue and pain.

Methodology

Perhaps it could be better explained that these are 2 different studies with different inclusion and exclusion criteria at the beginning of this section.

Regarding the first part: Was the sample contacted via online previously diagnosed or after invitation to the study?

In the creation of the questionnaire, was there no physiotherapist?

Was the sample contacted by social media how? Advertisements, was it informed from the general practitioner or associations?

No data was recorded on the limitations caused by fibromyalgia or symptoms presented by these participants?

Who distributed the questionnaires, who collected and analyzed the data, were there any missing or lost data?

Part 2: Who selected the focus group, why were these barriers selected, was there a prior question as to which barrier was most likely to be encountered, was there a prior question as to which barrier was most likely to be encountered?

It would be convenient to add a graph with the flow diagram of the sample.

Results

As an appreciation, although it is not the objective of the study, were aspects of the sample that are happy or do not report difficulty in the work (14%) analyzed or taken into account?

Is there any information on pharmacological or non-pharmacological treatments?

Discussion

Up to this section nothing has been mentioned in relation to other publications dealing with furniture adaptations.

Conclusion

The conclusions are very general and should be adapted more specifically to the proposed objectives and results obtained.

Author Response

Reviewer 3

The manuscript describes a quantitative and qualitative assessment of patients suffering from fibromyalgia in relation to their work environment.

Introduction

  1. Well referenced but perhaps some mention could be made of previous studies that assess the interventions performed for these patients in the work environment, as well as the other symptoms that these patients also report besides fatigue and pain.

ANSWER: Following the reviewer’s suggestion, other FM symptoms and comorbidities have now been described in the Introduction (lines 38-43). For what concerns interventions in the workplace, to the best of our knowledge, and according with Annie Palstam* and Kaisa Mannerkorpi (PMID: 28464770), studies evaluating work disability as outcome in FM are scarce. A recent Randomized Controlled Trial by Haugmark and colleagues (PMID: 34187823) investigated the effects of a multicomponent rehabilitation programme, compared to treatment as usual, for FM patients on different outcomes including work ability. This study found a slight reduction in work impairment from baseline to 12-month follow-up in both experimental and control group, without significant differences. However, studies like this are not specifically developed to improve work ability or workplace wellbeing and we do not know any studies with this aim. A simple search on PubMed with the keywords “fibromyalgia” AND “workplace intervention” returns only 39 results, none of which represents a real workplace intervention.

Methodology

  1. Perhaps it could be better explained that these are 2 different studies with different inclusion and exclusion criteria at the beginning of this section.

ANSWER: Thank you for this suggestion that allowed us to better clarify the structure of the study. We have added these specifications at the beginning of the section (see lines: 96-97).

  1. Regarding the first part: Was the sample contacted via online previously diagnosed or after invitation to the study?

ANSWER: The potential sample of participants for the cross-sectional survey was invited online, it was an anonymous sample. An invitation was sent by means of the Facebook page and the website of the CFU that is an association of fibromyalgia patients. Every patient/member of CFU could see the invitation by entering the Facebook page or the website, and after reading about aims, procedures and burden of the survey, they could decide whether to participate or not. Considering this approach, participants had already received a diagnosis of FM before the survey, and they self-reported it. We made the whole process of recruitment clearer, and we specifically improved the Methods sections (see lines 108-115).

  1. In the creation of the questionnaire, was there no physiotherapist?

ANSWER: No physiotherapists were involved in the creation of the questionnaire. For clearness, the composition of the group who designed the questionnaire is specified at lines: 117-121.

  1. Was the sample contacted by social media how? Advertisements, was it informed from the general practitioner or associations?

ANSWER: The sample was contacted through the Facebook page and the website of the Association of FM Patients, CFU-Italia, while GPs were not involved. The procedures are specified in the manuscript (see lines indicated for point n. 3 above).

  1. No data was recorded on the limitations caused by fibromyalgia or symptoms presented by these participants?

ANSWER: No data on FM impact were collected, mainly to reduce the number of items within the online survey and to decrease respondents’ burden, thus favoring the greatest participation.

  1. Who distributed the questionnaires, who collected and analyzed the data, were there any missing or lost data?

ANSWER: The questionnaire was not physically distributed, because the survey was conducted online, using Microsoft forms (see also answer to points 3 and 5, above). The survey data were analyzed by Riccardo Vincis, Michael Tenti, Corrado Fagnani, and Laura Camoni. The Focus Group data were analyzed by Michael Tenti, Mery Paroli, and Virgilia Toccaceli (see also “Author contributions”). For what concerns missing data, we have now better specified that 41 subjects were excluded because they did not answer the items on age and/or on time from FM diagnosis. A flow-chart has been added to the manuscript (see new Figure 1). We also specified in the manuscript the number of missing data for “employment status” (see line 186 and line 189) and that from the questions on “Attitudes and difficulties experienced at work” (Section 3.1.2.) onwards, the number of respondents decreases to 1163 (see line 190).

  1. Part 2: Who selected the focus group, why were these barriers selected, was there a prior question as to which barrier was most likely to be encountered?

ANSWER: As specified in the “Materials and Methods” section (lines 140-142 and new Figure 1), the recruitment of subjects for the Focus Group (FG) study started through the launch of dedicated posts on the Facebook page and groups of CFU-Italia Odv, addressing FM people who had declared, participating in the previous survey, to have experienced difficulties in the work environment. Thirty-two FM sufferers communicated their interest in participating via email and fifteen were admitted after an interview with a clinical psychologist (Michael Tenti) who verified satisfaction of the eligibility criteria. This is now better specified at lines 144-145. For what concerns the survey questionnaire, we have included two open questions on perceived barriers and facilitators in the work environment. However, answers were high heterogeneous and somewhat confusing (they sounded as emotional complaints). Therefore, the FG had the aim of clarifying the contents of these open answers (see lines 131-134). A first question that was proposed in the FG investigated how to improve work environment in general, and in this phase, the participants were free to introduce their experience of the barriers; while a subsequent question investigated the specific barrier of “not being believed”, as it was the most recurring in the open questions posed during the online survey. No specific prior questions as to which barrier was most likely to be encountered was included in the FG study. However, after a brief presentation of the survey preliminary results, the coherence between the FG participants experience with the survey findings was investigated by means of a transition question (see Supplementary materials) that introduced to the real FG discussion.

  1. It would be convenient to add a graph with the flow diagram of the sample.

ANSWER: A flow-chart has now been introduced to better describe recruitment, eligibility of the samples as well as the final number of questionnaires (for the online survey) analyzed and  the number of participants selected for the two FGs. We particularly thank the reviewer for this suggestion, which allowed us to improve the description of the whole study methodology and procedures (see new Fig. 1 “Flow chart of recruitment for the two studies” in the Materials and Methods section).

Results

  1. As an appreciation, although it is not the objective of the study, were aspects of the sample that are happy or do not report difficulty in the work (14%) analyzed or taken into account?

ANSWER: We thank the reviewer for this comment. We have reported that some people were happy to go to work or declared no particular difficulties at work, and we think it would be interesting to explore the reasons behind this observation. We did not deepen this result in our study. However, we call for future studies to address more precisely those factors favoring   wellbeing at work of people with fibromyalgia.

  1. Is there any information on pharmacological or non-pharmacological treatments?

ANSWER: No data on pharmacological or non-pharmacological treatments were collected for the same reasons why information on the impact of fibromyalgia (see point. N. 6) was also not collected (i.e., to decrease respondents’ burden and favor the greatest possible participation).

Discussion

  1. Up to this section nothing has been mentioned in relation to other publications dealing with furniture adaptations.

ANSWER: A new sub-heading “Clinical implications” has now been introduced in Discussion. Within this sub-section, a focus on the state of the art of the research on ergonomics interventions and furniture adaptations for FM subjects has been developed, addressing a few researches from 2010 up to now (see lines 456-472).

Conclusion

  1. The conclusions are very general and should be adapted more specifically to the proposed objectives and results obtained.

ANSWER: We thank the reviewer for this comment. Conclusions are now better focused on the specific results obtained.

Round 2

Reviewer 1 Report

Comments and Suggestions for Authors

The current version of your manuscript has improved over the previous one, so I recommend its publication.

Congratulations

Comments on the Quality of English Language

No comments

Reviewer 3 Report

Comments and Suggestions for Authors

Dear authors,

Thank you for considering my suggestions.

Manuscript is clearer for me.